# Simulated Microgravity Alters P-Glycoprotein Efflux Function and Expression via the Wnt/β-Catenin Signaling Pathway in Rat Intestine and Brain

**DOI:** 10.3390/ijms24065438

**Published:** 2023-03-12

**Authors:** Huayan Liu, Min Liang, Yulin Deng, Yujuan Li

**Affiliations:** School of Life Science, Beijing Institute of Technology, No.5 Zhongguancun South Street, Haidian District, Beijing 100081, China

**Keywords:** permeability glycoprotein, efflux function, simulated microgravity, intestinal absorption, brain distribution, Wnt/β-catenin signaling pathway

## Abstract

The drug efflux transporter permeability glycoprotein (P-gp) plays an important role in oral drug absorption and distribution. Under microgravity (MG), the changes in P-gp efflux function may alter the efficacy of oral drugs or lead to unexpected effects. Oral drugs are currently used to protect and treat multisystem physiological damage caused by MG; whether P-gp efflux function changes under MG remains unclear. This study aimed to investigate the alteration of P-gp efflux function, expression, and potential signaling pathway in rats and cells under different simulated MG (SMG) duration. The altered P-gp efflux function was verified by the in vivo intestinal perfusion and the brain distribution of P-gp substrate drugs. Results showed that the efflux function of P-gp was inhibited in the 7 and 21 day SMG-treated rat intestine and brain and 72 h SMG-treated human colon adenocarcinoma cells and human cerebral microvascular endothelial cells. P-gp protein and gene expression levels were continually down-regulated in rat intestine and up-regulated in rat brain by SMG. P-gp expression was regulated by the Wnt/β-catenin signaling pathway under SMG, verified by a pathway-specific agonist and inhibitor. The elevated intestinal absorption and brain distribution of acetaminophen levels also confirmed the inhibited P-gp efflux function in rat intestine and brain under SMG. This study revealed that SMG alters the efflux function of P-gp and regulates the Wnt/β-catenin signaling pathway in the intestine and the brain. These findings may be helpful in guiding the use of P-gp substrate drugs during spaceflight.

## 1. Introduction

During spaceflight, astronauts face extremely complex space environments, such as microgravity (MG), radiation, and noise [1]. MG is an adverse factor that is continuous and cannot be separated from spaceflight. Exposure to this condition could lead to cardiovascular dysfunction [2], muscle atrophy [3], bone loss [4], nausea and vomiting [5], and immune [6] and intestinal epithelial barrier function deficiency [7]. Oral drugs are used to prevent or treat the multisystem physiological damage induced by MG to ensure the health and safety of astronauts [8,9]. Several efflux transporters located in intestinal epithelial cells (IECs) and brain microvascular endothelial cells, such as permeability glycoprotein (P-gp), are involved in the absorption and distribution of oral drugs [10,11].

P-gp is a 170 kD transmembrane glycoprotein and a member of the ATP-binding cassette superfamily [12]. It is encoded by the *MDR1* gene in humans and homologs *mdr1a* and *mdr1b* genes in rodents [13], and its expression is regulated by multiple signaling pathways, including the Wnt/β-catenin signaling pathway [14,15,16]. P-gp is abundantly found in the luminal membrane of the small intestine and blood-brain barrier (BBB) and in the apical membrane of excretory cells, such as hepatocytes and renal proximal tubular epithelium [17]. Owing to its strategic location, P-gp can functionally limit the cellular transport of drugs from the gastrointestinal lumen into enterocytes and from the blood circulation into the brain [18]. Therefore, P-gp plays an important role in the intestinal absorption and brain distribution of its substrate drugs. Several drugs used by astronauts during spaceflight are substrates of P-gp, such as acetaminophen (AP) [19], ciprofloxacin [20], and ibuprofen [21]. At present, astronauts use medications in accordance with terrestrial medical practices; however, whether the absorption or distribution of drugs during spaceflight is the same as on Earth remains unclear [8]. Our previous studies showed that the expression of P-gp decreased in rat ileum mucosa and increased in rat brain under simulated MG (SMG) [22,23]. However, the MG-induced change in the efflux function of P-gp and the potential mechanism have not been elucidated. If the efflux function of P-gp in the intestine and brain is altered under MG, these changes may affect the intestinal absorption and brain distribution of P-gp substrate drugs [24]. Any changes in absorption or distribution or both may lead to the change of efficacy or toxicity of the drug [25]. Therefore, studying the changes in P-gp efflux function and expression in the intestine and brain under MG is important to provide a new perspective for the rational use of P-gp substrate drugs during spaceflight. The Wnt signaling pathway is quite sensitive when in space, and 18 days of spaceflight significantly affected this pathway in melanoma cells [26]. Our previous study also found that the expression of β-catenin was altered in the rat intestinal mucosa and the rat brain under SMG [22,27]. Whether SMG regulates P-gp expression in rat small intestine and rat brain through the Wnt/β-catenin signaling pathway has not been reported.

In this study, tail-suspended rat, random 3D rotary cultured human colon adenocarcinoma cell (Caco-2), and human cerebral microvascular endothelial cell (hCMEC/D3) models were used to simulate MG. The effect of different SMG duration on P-gp efflux function and expression in rat intestine and brain was elucidated. Whether SMG regulates P-gp expression in the intestine and brain through the Wnt/β-catenin signaling pathway was also explored and validated using pathway-specific agonist and inhibitor. Furthermore, the intestinal absorption and brain distribution of AP were determined to verify the change in P-gp efflux function. This study gained insights into the effects of SMG on P-gp efflux function and expression, explored its regulation mechanism, and provided scientific support as guidelines for the use of P-gp substrate drugs during spaceflight.

## 2. Results

### 2.1. Smg Inhibited the Efflux Function and Expression of P-Gp in Rat Intestine and Caco-2 Cells

#### 2.1.1. SMG Inhibited the Efflux Function and Expression of P-Gp in Rat Intestine

The cyclosporine A (CsA) level in the rat plasma was used to reflect the P-gp efflux function in rat intestine [28]. The CsA level in rat plasma was increased by 16.1% and 49.0% after 7 and 21 days of SMG treatment, respectively (Figure 1A). This finding was similar to the results after the oral administration of Verapamil (Ver), a typical inhibitor of P-gp efflux function. These results indicated that 7 and 21 days of SMG treatment significantly inhibited the efflux function of P-gp in rat intestine.

Western-Blot analysis indicated that the expression of P-gp decreased by 38.4% and 56.8% after 7 and 21 days of SMG treatment, respectively (Figure 1B–D). The relative expression levels of P-gp were expressed as the ratio of the gray value of the P-gp band to that of the total proteins in the same lane. The total protein gels are shown in Appendix A: immunohistochemical (IHC) assay revealed that the P-gp staining area was significantly reduced after 7 and 21 days of SMG treatment (Figure 1E,F), and the expression of P-gp was down-regulated in rat intestine. Quantitative PCR (qPCR) showed that the expression levels of *mdr1a* and *mdr1b* in rat intestine were also remarkably down-regulated in 7 and 21 days of SMG-treated rats (Figure 1G). Therefore, 7 and 21 days of SMG treatment time dependently inhibited the efflux function and expression of P-gp in rat intestine.

#### 2.1.2. 72 h SMG Treatment Inhibited the Efflux Function and Expression of P-Gp in Caco-2 Cells

As reflected by the intracellular rhodamine 123 (Rho123) fluorescence intensity (Figure 2A), 72 h of SMG exposure significantly inhibited the function of P-gp in the Caco-2 cells. The protein expression of P-gp was consistently down-regulated after 12, 24, 48, and 72 h of SMG exposure (Figure 2B,C). The total protein gels are shown in Appendix A. Immunofluorescence results showed that 12, 24, 48 and 72 h of SMG treatment time dependently down-regulated the expression of P-gp in the Caco-2 cells (Figure 2D,E). qPCR revealed that the expression of *MDR1* was also consistently down-regulated by SMG treatment (Figure 2F). In summary, 72 h of SMG exposure significantly inhibited the efflux function and expression of P-gp in the Caco-2 cells. However, 12, 24, and 48 h of SMG exposure promoted the efflux function of P-gp while inhibiting its expression in Caco-2 cells.

### 2.2. SMG Inhibited the Efflux Function and Activated the Expression of P-Gp in Rat Brain and hCMEC/D3 Cells

#### 2.2.1. SMG Inhibited the Efflux Function and Activated the Expression of P-Gp in Rat Brain

The efflux function of P-gp in rat brain was reflected by the ratio of Rho123 concentration in rat brain and blood. After 7 and 21 days of SMG treatment, the ratio of Rho123 concentration in rat brain to that in the blood increased by 40.5% and 173.8%, respectively, compared with that in the control (CON) group (Figure 3A). These results indicated that 7 and 21 days of SMG treatment significantly inhibited the efflux function of P-gp in rat brain. Western-Blot results showed that the expression of P-gp increased by 23.4% and 24.2% after 7 and 21 days of SMG treatment, respectively (Figure 3B–D). The total protein gels are shown in Appendix A. IHC results also demonstrated the increase in P-gp expression after 7 and 21 days of SMG exposure (Figure 3E,F), and these findings were in accordance with the Western-Blot results. qPCR displayed that *mdr1a* gene expression increased by 28.5% and 57.5% in the brain of 7 and 21 days SMG-treated rats (Figure 3G). In conclusion, 7 and 21 days of SMG treatment could significantly inhibit the efflux function of P-gp in rat brain while promoting its expression in a time-dependent manner.

#### 2.2.2. 48 and 72 h SMG Treatment Inhibited the Efflux Function and Activated the Expression of P-Gp in hCMEC/D3 Cells

Figure 4A shows that 48 and 72 h of SMG treatment inhibited the efflux function of P-gp in hCMEC/D3 cells, resulting in a 13.4% and 16.7% increase in the fluorescence intensity of intracellular Rho123, respectively. Western-Blot analysis showed that P-gp expression was up-regulated by approximately 38.3%, 16.4%, and 81.9% after 24, 48, and 72 h of SMG treatment, respectively (Figure 4B,C). The total protein gels are shown in Appendix A. Immunofluorescence results also showed that P-gp expression increased after SMG treatment (Figure 2D,E). Finally, qPCR results showed that *MDR1* gene expression level increased after 24 and 48 h of SMG treatment but decreased by 82.1% after 72 h of SMG treatment. In summary, 48 and 72 h of SMG treatment significantly inhibited the efflux function and protein expression of P-gp in hCMEC/D3 cells. However, 24 h of SMG treatment promoted the efflux function of P-gp while inhibiting its protein and gene expression in hCMEC/D3 cells.

### 2.3. SMG Down-Regulated P-Gp Expression by Inhibiting the Wnt/β-Catenin Signaling Pathway in Rat Intestine and Caco-2 Cells

The expression levels of Wnt3a, phosphorylated disheveled 2 (pho-Dvl2), glycogen synthase kinase-3β (GSK-3β), pho-GSK-3β, β-catenin, and Dickkopf 1 (DKK1) were determined using Western-Blot. Figure 5A–C shows that after 7 and 21 days of SMG treatment, the expression of Wnt3a decreased without significant difference, and that of pho-Dvl2, pho-GSK-3β/GSK-3β, and β-catenin was dramatically down-regulated. The total protein gels are shown in Appendix A. This finding indicated that the Wnt/β-catenin pathway was inhibited by SMG. Meanwhile, the expression of DKK1, an inhibitor of the Wnt/β-catenin pathway, increased significantly. On the basis of the aforementioned results, both 7 and 21 days of SMG treatment could promote DKK1 expression and inhibit the Wnt/β-catenin signaling pathway in rat intestine.

Figure 5D,E shows that the expression of Wnt3a and pho-GSK-3β/GSK-3β was down-regulated after 12, 24, 48, and 72 h of SMG treatment, but the difference was significant only after 72 h of SMG treatment. The expression of pho-Dvl2 and β-catenin was down-regulated after SMG treatment, and that of DKK1 showed an increasing trend after 12, 24, 48, and 72 h of SMG treatment. This finding indicated that SMG also promoted DKK1 expression and inhibited the Wnt/β-catenin signaling pathway in Caco-2 cells in a time-dependent manner.

To explore whether the SMG-induced change in intestinal P-gp expression is regulated by the Wnt/β-catenin signaling pathway, Caco-2 cells were treated with 2 μM FzM1.8 (an agonist of the Wnt/β-catenin signaling pathway) [29] before 48 h of SMG treatment. Figure 5F,G shows that compared with those in the CON group, the Wnt/β-catenin signaling pathway was activated, and P-gp expression was increased in the Caco-2 cells after 48 h of FzM1.8 treatment. After 48 h of SMG treatment, the Wnt/β-catenin signaling pathway of Caco-2 cells was inhibited, and P-gp expression was significantly decreased. Meanwhile, FzM1.8 treatment could counteract the effect of SMG and up-regulate P-gp expression. In summary, 48 h of SMG treatment might down-regulate the expression of P-gp in Caco-2 cells via the Wnt/β-catenin signaling pathway.

### 2.4. SMG Up-Regulated P-Gp Expression by Activating the Wnt/β-Catenin Signaling Pathway in Rat Brain and hCMEC/D3 Cells

The expression levels of Wnt3, pho-Dvl2, GSK-3β, pho-GSK-3β, and β-catenin in rat brain were determined using Western-Blot. The total protein gels are shown in Appendix A. Figure 6A–C shows that the expression levels of pho-Dvl2 and pho-GSK-3β/GSK-3β tended to increase without significant difference, and those of Wnt3 and β-catenin were dramatically up-regulated after 7 days of SMG treatment. After 21 days of SMG treatment, the expression levels of Wnt3, pho-Dvl2, pho-GSK-3β/GSK-3β, and β-catenin were all significantly increased, indicating that the Wnt/β-catenin pathway in rat brain was activated by 21 days of SMG treatment.

As shown in Figure 6D,E, the expression levels of Wnt3, pho-Dvl2, and β-catenin were significantly decreased after 24 h of SMG treatment. The ratio of pho-GSK-3β to GSK-3β also tended to be down-regulated by 24 h of SMG treatment. Although no obvious change was observed in the expression of Wnt3 under 48 h of SMG treatment, the expression levels of pho-Dvl2, pho-GSK-3β/GSK-3β, and β-catenin were all significantly increased. Furthermore, after 72 h of SMG treatment, the expression of Wnt/β-catenin signaling pathway proteins in hCMEC/D3 cells did not change significantly, except for the decrease in pho-Dvl2 expression.

Under 48 h of SMG treatment, the change trend of P-gp expression was consistent with that of the Wnt/β-catenin signaling pathway in the hCMEC/D3 cells. Hence, the hCMEC/D3 cells were treated with 1 nM IWP-O1 (an inhibitor of the Wnt/β-catenin signaling pathway) [30] before 48 h of SMG treatment. Western-Blot assay results showed that compared with those in the CON group, the Wnt/β-catenin signaling pathway was inhibited, and P-gp expression was decreased in the hCMEC/D3 cells after IWP-O1 treatment for 48 h (Figure 6F,G). After 48 h of SMG treatment, the Wnt/β-catenin signaling pathway of hCMEC/D3 cells was activated, and P-gp expression significantly increased. Meanwhile, IWP-O1 treatment could reverse the effect of 48 h SMG treatment and down-regulate P-gp expression. In summary, 48 h of SMG treatment might up-regulate the expression of P-gp in hCMEC/D3 cells via the Wnt/β-catenin signaling pathway.

### 2.5. SMG Promoted the In Vivo Intestinal Absorption and Brain Distribution of AP

The absorption of AP in the rat ileum was determined using single-pass intestinal perfusion (SPIP), and its penetration was evaluated according to the amount of luminal disappearance. The effective permeability (*P*_eff_) of absorbed solutes in rats correlates well with estimated *P*_eff_ in humans, and the SPIP model can be used to predict in vivo absorption in humans and evaluate the specific contribution of drug transporters [31,32]. The contents of AP in importer and exporter perfusate were determined using fully validated high-performance liquid chromatography (HPLC)-UV. Retention times for AP and internal standard (IS; ferulic acid) were 7.9 and 10.5 min, and all peaks were well separated (Appendix A). The absorption constant (*K*_a_) and *P*_eff_ of AP were calculated using Equations (1) and (2), and the results are shown in Table 1. After 7 and 21 days of SMG treatment, the *K*_a_ of AP in rat ileum increased by 10.3% and 54.8%, respectively, and the *P*_eff_ increased by 14.1% and 74.2%, respectively. This finding was similar to the results after the oral administration of Ver. In summary, 7 and 21 days of SMG treatment promoted the absorption of AP in rat intestine. The increased in vivo intestinal absorption of AP also proved that P-gp efflux function in the intestine was inhibited under SMG.

The rat brain distribution of AP was reflected by the ratio of the AP concentration in rat brain (μg/g) to that in rat plasma (μg/g) at 2 h post-oral administration of AP. Figure 7 shows that the brain-to-plasma concentration ratio (*K*_p_ brain) of AP exhibited an increasing trend after 7 days of SMG treatment and reached approximately 22.0% higher than that in the CON group. Meanwhile, 21 days of SMG treatment significantly increased the brain-to-plasma concentration ratio of AP by 40.7% compared with that in the CON group. The results of AP brain distribution also proved that the P-gp efflux function was inhibited in rat brain after 7 and 21 days of SMG treatment. In summary, the elevated intestinal absorption and brain distribution levels of AP confirmed the inhibited P-gp efflux function in rat intestine and brain under SMG.

## 3. Discussion

During spaceflight, drugs are always used to reverse the physiologic insult induced by the complex space environment [33]. To date, these drugs are being administered under the assumption that they act as safely and efficaciously as they do on Earth. However, this assumption has not been systematically investigated [34]. If changes in the pharmacokinetics or pharmacodynamics of drugs taken during spaceflight are not fully considered, drug efficacy and safety will not be guaranteed [8]. The intestinal absorption of oral drugs could be mediated by several efflux transporters, including P-gp. The changes in P-gp efflux function in the IEC membrane might affect the amount of its substrate drugs absorbed via the intestine. The differences in efflux function or expression, or both, of intestinal P-gp potentially induce changes in drug bioavailability, efficacy, and safety [35,36,37]. This study first found that the efflux function and expression of P-gp were significantly inhibited in the 7 and 21 days of SMG-treated rat intestine and the 72 h of SMG-treated Caco-2 cells, implying that the intestinal absorption of P-gp substrate drugs might be promoted under SMG conditions. With prolonged SMG treatment, the efflux function and expression of P-gp in the intestine were significantly inhibited. The aforementioned results were validated by the SPIP analysis of AP, a substrate drug of P-gp [19]. Therefore, 7 and 21 days of SMG treatment could significantly promote AP intestinal absorption. Whether this effect alters the efficacy of AP or potentially causes toxicity warrants further investigation [38].

P-gp at the membranes of cerebral microvascular endothelial cells functions as an active efflux pump by extruding a wide range of substrates from the brain, including most drugs [39]. When central nervous system drugs cross the BBB, P-gp is an important factor limiting the drug delivery to the central nervous system and consequently reducing the efficacy of the drug [40]. Inhibiting the efflux function of P-gp on BBB could be beneficial to some neurological disease drugs to exert their efficacy [41,42,43]. For example, the coadministration of phenytoin and a P-gp inhibitor was significantly more effective in controlling seizures than phenytoin administration alone [44,45]. The efflux function of P-gp in the brain could be reflected by the *K*_p_ brain of P-gp substrate [46,47]. In this study, the effect of SMG on P-gp efflux function in rat brain was reflected by the *K*_p_ brain of Rho123. The results showed that P-gp efflux function was inhibited in 7 and 21 days of SMG-treated rat brain and 48 h and 72 h of SMG-treated hCMEC/D3 cells. The *K*_p_ brain of AP in 7 and 21 days of SMG-treated rats was higher than that in the CON group, revealing the inhibited P-gp efflux function in rat brain. Many central analgesic drugs, including AP, are substrates of P-gp. When P-gp efflux function is inhibited, the efficacy of these drugs may be enhanced [48]. Therefore, the inhibition of P-gp efflux function in the brain under SMG conditions might facilitate central nervous system drugs to cross the BBB and exert their efficacy. Hence, the adverse reactions possibly caused by excessive accumulation of drugs in the brain also warrant research [49,50].

During spaceflight, astronauts face various physiological changes, such as fluid shifts, changes in local blood flow, drug-binding plasma protein levels, and altered gastrointestinal motility. These changes potentially affect drug pharmacodynamics or pharmacokinetics [51]. Some oral drugs taken during spaceflight do not exhibit the expected effect [52,53,54,55]. Studies conducted in orbit have evaluated the salivary pharmacokinetics of AP [56]. The plasma pharmacokinetics of AP under SMG conditions have also been investigated. The results revealed that the absorption of AP in two astronauts increased on mission day 2 compared with that before the flight [57]. Additionally, the absorption of AP was promoted by 21 and 28 days of SMG treatment in rats and 80 days of SMG treatment in humans [38,58]. In the present study, SPIP analysis in rats showed that 7 and 21 days of SMG treatment could significantly promote AP intestinal absorption in vivo. The increased intestinal absorption of AP might be due to the inhibition of intestinal P-gp efflux function under SMG. Additionally, the absorption of P-gp oral drug substrates ibuprofen, Ver, propranolol, and promethazine in humans was promoted under SMG [59,60,61]. This finding may also result from the inhibited P-gp efflux function in the intestine under SMG. The present study suggested that the changes of P-gp efflux in the intestine and brain under a microgravity environment may lead to the difference in intestinal absorption and brain distribution of P-gp substrate oral drugs compared with that on the ground. This indicates that the dosing regimen should be carefully considered when these P-gp substrate drugs are taken by astronauts, including administered dose, frequency, administration routes, or potential drug-drug interactions. The optimization of drug administration thus requires further in-orbit validation.

The Wnt/β-catenin signaling pathway can regulate the expression of P-gp [14,15,16]. The expression of REKEN cDNA 2210419D22 in melanoma cells in space was significantly different from that in Earth, and this phenomenon was related to the Wnt/β-catenin signaling pathway [26]. The canonical Wnt/β-catenin signaling pathway is also involved in the control of the response to SMG in nematode Caenorhabditis elegans [62]. Our previous study found that the expression of β-catenin was altered in rat intestinal mucosa and brain under SMG [22,27]. Therefore, the present study further investigated whether SMG regulates P-gp expression in the brain and intestine through the Wnt/β-catenin signaling pathway. The composition and regulation of the Wnt/β-catenin signaling pathway were as follows. In the absence of Wnt, β-catenin in the cytoplasm is phosphorylated by a degradation complex consisting of axis inhibitor (Axin), adenomatous polyposis coli protein (APC), casein kinase 1 (CK1), and GSK-3β, and is then further ubiquitinated and degraded by the ubiquitin–proteasome system [63]. Once Wnt binds to Frizzled (Fzd) and lipoprotein receptor-related proteins 5/6 (LRP5/6), the cytoplasmic Dvl is phosphorylated [64], the β-catenin degradation complex is disassembled, and GSK-3β is phosphorylated and binds to LRP5/6 [65,66], hence impeding β-catenin degradation. β-Catenin accumulates in the cytoplasm, enters the nucleus, and binds to transcription factor (TCF) to promote the expression of target genes, including those encoding P-gp [67,68,69]. GSK-3β phosphorylation, can also activate the Wnt/β-catenin signaling pathway [70,71].

This study found that the levels of Wnt3a, pho-Dvl2, pho-GSK-3β/GSK-3β, β-catenin, and P-gp in rat ileum were down-regulated by 7 and 21 days of SMG treatment. The Wnt/β-catenin signaling pathway was also inhibited by SMG treatment in a time-dependent manner. For further investigation on whether SMG regulates the expression of P-gp in the intestine via the Wnt/β-catenin signaling pathway, 48 h of SMG treatment was selected, and the pathway-specific agonist FzM1.8 was added for verification. The results showed that the addition of FzM1.8 could counteract the inhibitory effect of 48 h of SMG treatment on the Wnt/β-catenin signaling pathway and increase the expression of P-gp. Therefore, SMG may down-regulate P-gp expression by inhibiting the Wnt/β-catenin signaling pathway.

The Wnt/β-catenin signaling pathway was significantly activated in the brain of 7 and 21 days of SMG-treated rats and in 48 h of SMG-treated hCMEC/D3 cells. The addition of exclusive inhibitor IWP-O1 could attenuate the activation effect of 48 h of SMG treatment on the Wnt/β-catenin signaling pathway and decrease the expression of P-gp. This finding suggested that SMG might promote the Wnt/β-catenin signaling pathway to increase P-gp expression in the brain. However, after 24 and 72 h of SMG treatment, the change trend of P-gp expression in hCMEC/D3 cells was not consistent with that of the Wnt/β-catenin signaling pathway. In addition to the Wnt/β-catenin signaling pathway, various signaling pathways regulate P-gp expression, including protein kinase C [72], nuclear factor κB [73], mitogen-activated protein kinase [74], and PI3K/AKT signaling pathways [75]. Therefore, 24 and 72 h of SMG treatment may regulate the expression of P-gp in hCMEC/D3 cells through other pathways besides the Wnt/β-catenin signaling pathway; this finding warrants further research.

## 4. Materials and Methods

### 4.1. Animal Treatment and Sample Collection

Sprague–Dawley male, specific pathogen-free rats (200 ± 20 g) were obtained from the Institute of Laboratory Animal Science (Beijing, China). All animal experiments were approved by the Beijing Institute of Technology Animal Care and Use Committee (Protocol No-SYXK-BIT-20200109002, Beijing, China) and complied with the Guide for the Care and Use of Laboratory Animals (NIH publication No. 85–23, revised in 1996). All rats were housed in a controlled environment with a maintained 24 °C ± 1 °C and 55% ± 5% relative humidity and had free access to food and water. After 7 days acclimatization, the rats were divided into four groups (the *n* = 6 per group): 7 days-CON, 7 days-SMG, 21 days-CON, and 21 days-SMG. The rats in SMG groups were subjected to tail suspension and 30° head-down for 7 days or 21 days according to the Morey–Holton model [76].

At the end of 7 days or 21 days of SMG treatment, all rats were anesthetized via intraperitoneal injection with chloral hydrate (350 mg/kg), and the ileum was removed. Approximately 1 cm ileum was fixed in 4% paraformaldehyde (Solarbio, Beijing, China) over 24 h for IHC staining, and the remaining ileum was used to collect the mucosa. The rat brain was divided into two parts. The left brain was fixed in 4% paraformaldehyde for IHC staining. The right brain and ileum mucosa were stored at −80 °C for further experiments.

### 4.2. Efflux Function Analysis of P-Gp in Rat Intestine and Brain

After 7 days or 21 days of tail suspension, the CON rats were divided into the CON group and CON + Ver group (*n* = 6 per group). Ver (30 mg/kg; Solarbio, Beijing, China) was orally administered to the rats in the CON + Ver group, and CsA (15 mg/kg, TCI CO., LTD., Tokyo, Japan) was concurrently orally administered to all rats. After 3.5 h, all rats were anesthetized with chloral hydrate (350 mg/kg), and the plasma was sampled from the heart. The content of CsA in the rat plasma was determined using an ELISA kit following the kit instructions (Yuanmu Biotech Co., Ltd., Shanghai, China).

For the analysis of P-gp efflux function in rat brain, the rats in the CON + Ver group were injected with Ver (1 mg/kg) through the tail vein after being tail suspended for 7 days or 21 days. After 45 min, the tail veins of all rats were injected with Rho123 (0.5 mg/kg, Solarbio, Beijing, China). At 15 min post-injection, all rats were anesthetized with chloral hydrate (350 mg/kg), and the blood was collected from the heart. After being subjected to transcardiac perfusion using the stroke-physiological saline solution from the heart, the brain samples of all rats were collected. In summary, 0.1 g of brain samples were homogenized with 0.4 mL of normal saline to obtain brain homogenate. Blank plasma and brain samples were collected from rats without a Rho123 injection and used to prepare Rho123 standard curve samples. The concentration of Rho123 in plasma and brain homogenate from all rats was detected by fluorescence intensity (λ excitation/λ emission = 485/535) using a microplate reader (BioTek, Thorold, ON, Canada). The *K*_p_ brain of Rho123 was used to evaluate P-gp efflux function.

### 4.3. Immunohistochemical Staining

The rat ileum segments and brain fixed in 4% paraformaldehyde were embedded in paraffin and sectioned following routine procedures [77]. Slides were blocked in 1× Tris-buffered saline containing 3% bovine serum albumin (BSA), 2% serum, and 0.02% Tween 20 at room temperature for 30 min. The sections were incubated with the primary antibody against P-gp (1:400 for rat ileum and 1:100 for rat brain, Abcam, Cambridge, UK) overnight at 4 °C, followed by incubation with horseradish peroxidase (HRP)-conjugated goat anti-rabbit IgG (ZSGB-Bio, Beijing, China) at room temperature for 2 h. The sections were then visualized using a diaminobenzidine solution. Between each of the aforementioned steps, the sections were washed using 1× Tris-buffered saline tween (TBST) three times for 5 min each. Finally, images were captured using a Nanozoomer S210 microscopic-resolution scanner equipped with Digital Pathology View 2.0 software (Hamamatsu Photonics, Shizuoka, Japan). Java-based image-processing and analysis software (Image-Pro Plus; Version 6.0.0.260; National Institutes of Health, Bethesda, MD, USA) was used to analyze the proportion of P-gp-stained area in the whole ileal or cerebral cortex section of each rat in each group, respectively.

### 4.4. Cell Culture and SMG Treatment

Caco-2 cells were purchased from the Chinese Academy of Sciences (Shanghai, China) and maintained in MEM/EBSS medium (Hyclone, UT, USA) supplemented with 10% fetal bovine serum (Sofar, Beijing, China), 1% penicillin–streptomycin liquid (Solarbio, Beijing, China), and 1% nonessential amino acids (Gibco, New York, NY, USA) in a humidified 5% CO_2_ atmosphere at 37 °C. The Caco-2 cells were seeded in a T-12.5 flask (2 × 10^4^ cells/cm^2^) and cultured for 21 days under normal gravity. The T-12.5 flask was filled with culture medium while preventing bubble formation and then exposed to SMG using random speed 3D clinostat (National Space Science Center, Beijing, China) [78] for 12, 24, 48, and 72 h. The Caco-2 cells in the CON group were cultured under normal gravity in the same CO_2_ incubator.

The hCMEC/D3 cells were purchased from iCell Bioscience Inc. (Shanghai, China) and cultured in RPMI 1640 medium containing 1% penicillin–streptomycin liquid (Solarbio, Beijing, China) and 10% fetal bovine serum (Tianhang, Zhejiang, China) in the same environment as that of Caco-2 cells. The hCMEC/D3 cells were seeded at a density of 6 × 104 cells/cm^2^ and cultured for 2 days under normal gravity. The flasks were then filled with culture medium and exposed to SMG using random speed 3D clinostat for 24, 48, and 72 h. The hCMEC/D3 cells in the CON group were cultured under normal gravity in the same CO_2_ incubator.

FzM1.8 and IWP-O1 (Wnt/β-catenin signaling pathway-specific agonist and inhibitor, respectively) were dissolved in dimethyl sulfoxide (DMSO) and diluted with culture medium to obtain the desired concentrations. The final concentration of DMSO was 0.1%. FzM1.8 (2 μM) was added to the Caco-2 cells in the CON + FzM1.8 and SMG + FzM1.8 groups. IWP-O1 (1 nM) was added to the hCMEC/D3 cells in the CON + IWPO-1 and SMG + IWPO-1 groups. The cells in the SMG, SMG + FzM1.8, or SMG + IWPO-1 group were then exposed to SMG for 48 h.

### 4.5. Efflux Function Analysis of P-Gp in Caco-2 and hCMEC/D3 Cells

After 12, 24, 48, or 72 h of SMG treatment, the Caco-2 cells were treated with 2 mL of medium containing Rho123 (2 μM) for 1 h. The cells were then washed with phosphate buffer saline (PBS) solution five times (2 mL each time) and treated with 2 mL of 0.1% Triton X-100 for 15 min. The cell solution was transferred to a black 96-well plate (Corning, NY, USA) with six wells in each group and 200 μL in each well after brief centrifugation. The fluorescence intensity of each well was detected using a microplate reader (BioTek, Thorold, ON, Canada). The excitation/emission wavelength of Rho123 was 485/535 nm. After 24, 48, or 72 h of SMG treatment, the hCMEC/D3 cells were treated with 2 mL of medium containing Rho123 (2 μM) for 0.5 h. Subsequent operations were performed in the same way as for Caco-2 cells.

### 4.6. Western-Blot

Total proteins in the rat ileum mucosa and brain and Caco-2 and hCMEC/D3 cells were extracted using radio immunoprecipitation assay buffer containing protease inhibitors and protein phosphatase inhibitors. The supernatant was collected after centrifugation (12,000× *g*, 4 °C, and 10 min). The protein concentration was tested using Coomassie brilliant blue staining. The supernatant was mixed with 4× protein loading buffer containing dithiothreitol (Solarbio, Beijing, China) and desaturated in a boiling water bath for 10 min. The total protein (60 μg for ileum and brain and 30 μg for Caco-2 and hCMEC/D3 cells) of the samples was loaded in each well and separated by 12% sodium dodecyl sulfate–polyacrylamide gel electrophoresis using a TGX Stain-free^TM^ FastCast^TM^ Acrylamide Kit (Bio-Rad, Hercules, CA, USA). The total protein on the gel was imaged under ChemiDoc XRS+ mode of Image Lab software (version 3.0; Bio-Rad, Hercules, CA, USA) and then transferred to polyvinylidene fluoride membranes (Millipore, Burlington, MA, USA). The membranes were blocked with 5% non-fat milk or BSA for 2 h and incubated overnight at 4 °C with corresponding primary antibodies as follows: anti-P-gp (1:5000), anti-Wnt3a (1:1000), anti-pho-Dvl2 (1:5000), anti-β-catenin (1:5000), anti-DKK1 (1:500, Abcam, Cambridge, UK), anti-GSK-3β (1:2000), anti-pho-GSK-3β (1:2000, ZSGB-Bio, Beijing, China), and anti-Wnt3 (1:2000, Solarbio, Beijing, China). The membranes were then incubated with HRP-conjugated secondary antibody (ZSGB-Bio, Beijing, China) for 2 h at room temperature. Finally, the bands were visualized using an enhanced chemiluminescence reagent (Millipore, MA, USA). The gray value of each band was collected using Image Lab^TM^ Software (version 3.0, Bio-Rad, Hercules, CA, USA), and the relative expression levels of proteins were expressed as the ratio of the gray value of the target band to the total proteins in the same lane.

### 4.7. qPCR

Total RNA in the rat ileum mucosa and brain and Caco-2 and hCMEC/D3 cells was extracted using a total RNA extraction kit (Solarbio, Beijing, China) following the manufacturer’s instructions. The concentration and purity of RNA were determined by comparing the absorbance at 260 and 280 nm. In summary, 5 μg of total RNA was reverse transcribed to cDNA using a universal RT-PCR kit (M-MLV; Solarbio, Beijing, China). The relative expression level of P-gp mRNA to that of glyceraldehyde-3-phosphate dehydrogenase (GAPDH) was determined by qPCR using TB Green™ Premix Ex Taq™ II (TaKaRa, Tokyo, Japan) in accordance with the manufacturer’s instructions. The forward and reverse primers are shown in Table 2.

### 4.8. Immunofluorescence

The Caco-2 cells were cultured in a confocal plate for 10 days. The confocal plate was then filled with culture medium while ensuring no air bubbles had formed, and was then exposed to SMG using 3D clinostat for 12, 24, 48, and 72 h. The hCMEC/D3 cells were cultured on a four-well Lab-Tek II CC2 chamber slide overnight. The chamber slides were then filled with culture medium, sealed with parafilm while ensuring no air bubbles had formed, and exposed to SMG using 3D clinostat for 24, 48, or 72 h. After SMG treatment, the cells were washed with PBS three times for 5 min each, fixed with 4% paraformaldehyde for 30 min, and washed with PBS again three times. The cells were permeabilized with 0.5% Triton X-100 (Solarbio, Beijing, China) for 10 min, washed three times with PBS, blocked by 5% BSA for 2 h at room temperature, and incubated with rabbit anti-P-gp antibody (1:200) overnight at 4 °C. After being washed with PBS, the cells were incubated with Rho-conjugated goat anti-rabbit IgG (ZSGB-Bio, Beijing, China) for 2 h at room temperature and washed again. 0.1 μg/mL 4′,6-diamidino-2-phenylindole (Solarbio, Beijing, China) was then used to stain the nuclei for 15 min. After repeated washes in the same manner, the confocal plate or chamber slide was blow-dried and covered with an antifade solution (Solarbio, Beijing, China). Images were captured using Nikon N-SIM Confocal (Nikon, Tokyo, Japan) and NIS element imaging software (version 4.50). The results were analyzed using ImageJ, and the average fluorescence intensity on random lines was used to define P-gp expression in the cells.

### 4.9. Rat Single-Pass Intestinal Perfusion of Acetaminophen

Ver (30 mg/kg) was orally administered to the rats in the CON + Ver group. At 1 h post-oral administration, all rats were anesthetized with urethane (1.75 g/kg) and placed in a supine position. Approximately 10 cm of ileum was separated from the intestinal segments. Tubes were carefully inserted at both ends of this segment and ligated with a sterile surgical line. The segment was rinsed with 37 °C Krebs–Ringer’s (KR) solution, balanced with KR solution (containing 2.5 mg/mL AP) at a flow rate of 0.2 mL/min for 15 min, and continuously perfused with KR solution (containing 2.5 mg/mL AP). The flow rate was set as 0.2 mL/min. The outflow perfusate was collected in a pre-weighed tube, which was changed quickly every 15 min and weighed. Six tubes of this fluid were collected for each rat. The perfusion solution and outflow perfusate were diluted in methanol–water (1:9, *v*:*v*) and analyzed using HPLC. The absorption constant (*K*_a_) and effective permeability (*P*_eff_) were calculated using the following equations:(1)$$K_{a}=\frac{Q\left(1-C_{\mathrm{out}}V_{\mathrm{out}}/C_{\mathrm{in}}V_{\mathrm{in}}\right)}{{\pi r}^{2}L}$$
(2)$$P_{\mathrm{eff}}=\frac{-\mathrm{Qln}\left(C_{\mathrm{out}}V_{\mathrm{out}}/C_{\mathrm{in}}V_{\mathrm{in}}\right)}{2\pi \mathrm{rL}}$$
where Q is the perfusion flow rate; C_in_ and C_out_ are the inlet and outlet concentrations, respectively; V_in_ and V_out_ are the volumes of the importer and exporter perfusate, respectively; r is the internal diameter of the perfused intestinal segment (r = 0.18 cm) [79]; and L is the length of the perfused intestinal segment.

### 4.10. Rat Brain Distribution of Acetaminophen

After 7 or 21 days of tail suspension, all rats were orally administered AP (1.2 g/kg). At 1 h post-oral administration, all rats were anesthetized via intraperitoneal injection with chloral hydrate (350 mg/kg), and rat blood was taken from the heart. Rat brains were rapidly removed after perfusion with prechilled saline from rat hearts. Then, 0.1 mL of plasma from each rat was used to precipitate proteins with 300 μL of methanol. Approximately 0.1 g of the brain tissue of each rat was homogenized and ultrasonicated in 1 mL of methanol. The supernatant was removed and dried via vacuum centrifugation and reconstituted with methanol for HPLC. The brain distribution of AP was evaluated as the ratio of AP content in brain tissue (μg/g) to that in plasma (μg/mL).

### 4.11. Acetaminophen Determination Using the HPLC-UV Method

HPLC was performed on a Shimadzu LC-20AT HPLC system equipped with a UV detector (Shimadzu, Kyoto, Japan). Sample separation was conducted in a C18 column (5 μm, 4.6 × 150 mm). The mobile phases were methanol (A) and water (B) (1:9, *v*:*v*), and the flow rate was 1 mL/min. The volume of each sample injected into the HPLC system was 10 μL. The column temperature was maintained at 25 °C, and the detection wavelength was 275 nm. Ferulic acid (final concentration of 100 μg/mL) was used as an IS.

### 4.12. Statistical Analysis

Statistical analysis was performed using SPSS 20.0 software (IBM, New York, NY, USA), and the results were expressed as mean ± SD. The difference between the two groups was determined by one-way ANOVA, and a *p*-value less than 0.05 was considered to have statistical significance.

## 5. Conclusions

This study first elucidated that P-gp efflux function in rat intestine and brain was down-regulated by SMG. The efflux function of P-gp in Caco-2 and hCMEC/D3 cells was also inhibited by 72 h SMG treatment. Under SMG, P-gp expression decreased in rat intestine and Caco-2 cells and increased in rat brain and hCMEC/D3 cells. SMG might regulate P-gp expression in the intestine and brain via the Wnt/β-catenin signaling pathway as verified using the pathway-specific agonist in Caco-2 cells and inhibitor in hCMEC/D3 cells. The increased intestinal absorption and brain distribution levels of the P-gp substrate AP also confirmed the inhibition of P-gp function in the intestine and brain under SMG. These results might be helpful in understanding the effects and mechanism of SMG on P-gp function and expression in the intestine and brain, and provide scientific support as guidelines for the use of P-gp substrate drugs during spaceflight.

## Figures and Tables

**Figure 1 ijms-24-05438-f001:**
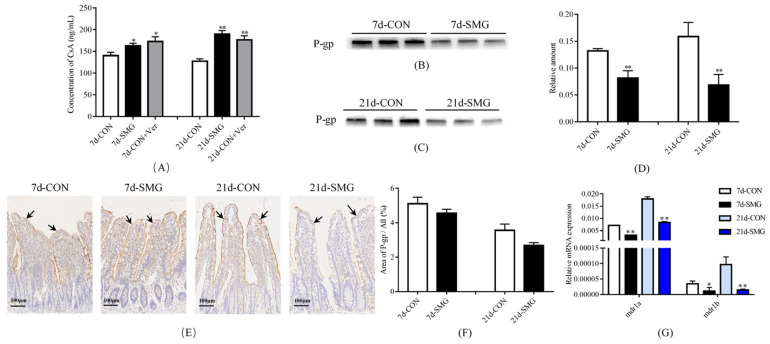
Simulated microgravity effect inhibited the function and expression of P-gp in rat intestines. (**A**) Concentration of CsA across intestinal epithelial cells to rats’ blood (*n* = 6). (**B**–**D**) The expression of P-gp in rat ileum mucosa determined using Western-Blot (*n* = 6). The relative expression levels of P-gp were expressed as the ratio of the gray value of the P-gp band to that of the total proteins in the same lane. The total protein gels are shown in Appendix A. (**E**,**F**), Immunohistochemistry (IHC) was performed to detect the expression of P-gp (arrows indicate the expression of P-gp) in rat ileum mucosa (*n* = 6). (**G**) mRNA expression of *mdr1a* and *mdr1b* in rat ileum mucosa detected using real-time qPCR (*n* = 6). * *p* < 0.05 and ** *p* < 0.01 vs. the CON group.

**Figure 2 ijms-24-05438-f002:**
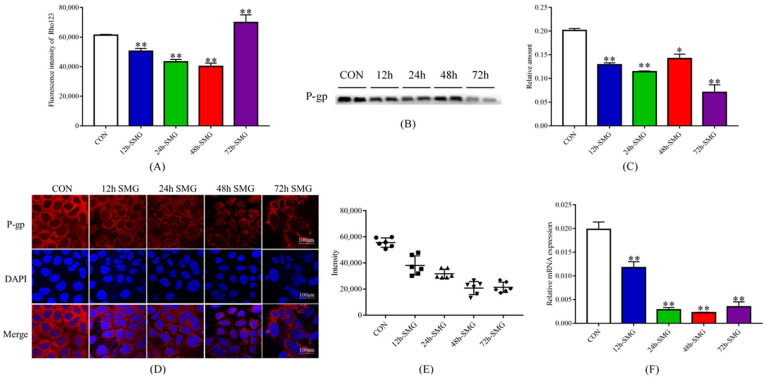
Simulated microgravity effect altered the function and expression of P-gp in Caco-2 cells. (**A**) Fluorescence intensity of Rho123 in Caco-2 cells. (**B**,**C**) The expression of P-gp in Caco-2 cells determined using Western-Blot. The relative expression levels of P-gp were expressed as the ratio of the gray value of the P-gp band to that of the total proteins in the same lane. The total protein gels are shown in Appendix A. (**D**,**E**) Immunofluorescence was performed to detect the expression of P-gp in Caco-2 cells. (**F**) mRNA expression of *MDR1* in Caco-2 cells detected using real-time qPCR. * *p* < 0.05 and ** *p* < 0.01 vs. the CON group.

**Figure 3 ijms-24-05438-f003:**
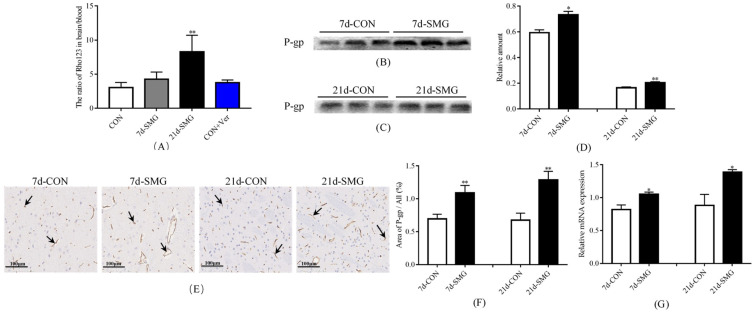
Simulated microgravity effect inhibited the function and activated expression of P-gp in rat brain. (**A**) Ratio of Rho123 concentration in rat brain and blood (*n* = 6). (**B**–**D**) The expression of P-gp in rat brain determined using Western-Blot (*n* = 6). The relative expression levels of P-gp were expressed as the ratio of the gray value of the P-gp band to that of the total proteins in the same lane. The total protein gels are shown in Appendix A. (**E**,**F**) Immunohistochemistry (IHC) was performed to detect the expression of P-gp (arrows indicate the expression of P-gp) in rat brain (*n* = 6). (**G**) mRNA expression of *mdr1a* in rat ileum mucosa detected using real-time qPCR (*n* = 6). * *p* < 0.05 and ** *p* < 0.01 vs. the CON group.

**Figure 4 ijms-24-05438-f004:**
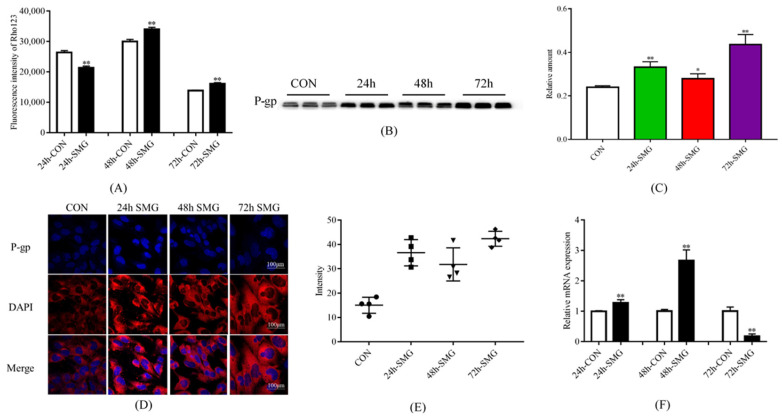
Simulated microgravity effect altered the function and expression of P-gp in hCMEC/D3 cells. (**A**) Fluorescence intensity of Rho123 in hCMEC/D3 cells. (**B**,**C**) The expression of P-gp in hCMEC/D3 cells determined using Western-Blot. The relative expression levels of P-gp were expressed as the ratio of the gray value of the P-gp band to that of the total proteins in the same lane. The total protein gels are shown in Appendix A. (**D**,**E**) Immunofluorescence was performed to detect the expression of P-gp in hCMEC/D3 cells. (**F**) mRNA expression of *MDR1* in hCMEC/D3 cells detected using real-time qPCR. * *p* < 0.05 and ** *p* < 0.01 vs. the CON group.

**Figure 5 ijms-24-05438-f005:**
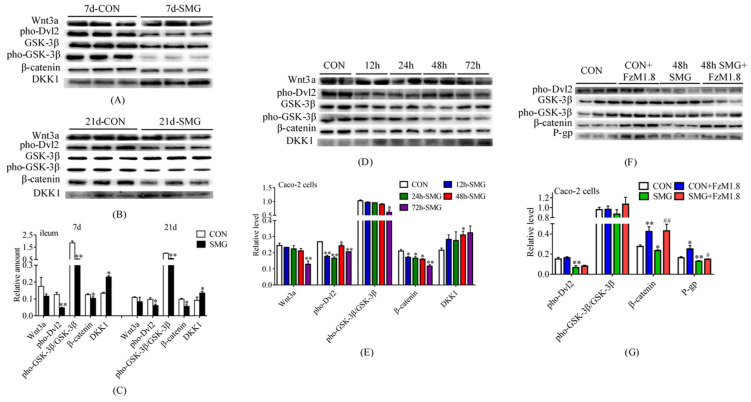
Simulated microgravity down-regulated the expression of P-gp by inhibiting the Wnt/β-catenin signaling pathway in rat intestine and Caco-2 cells. (**A**–**C**) Expression levels of Wnt3a, pho-Dvl2, GSK-3β, pho-GSK-3β, β-catenin, and DKK1 in rat ileum determined using Western-Blot. (**D**–**G**) Expression levels of Wnt3a, pho-Dvl2, GSK-3β, pho-GSK-3β, β-catenin, DKK1, and P-gp in Caco-2 cells determined using Western-Blot. The relative expression levels of proteins were expressed as the ratio of the gray value of the protein band to that of the total proteins in the same lane. The total protein gels are shown in Appendix A. * *p* < 0.05 and ** *p* < 0.01 vs. the CON group. # *p* < 0.05 and ## *p* < 0.01 vs. the SMG group.

**Figure 6 ijms-24-05438-f006:**
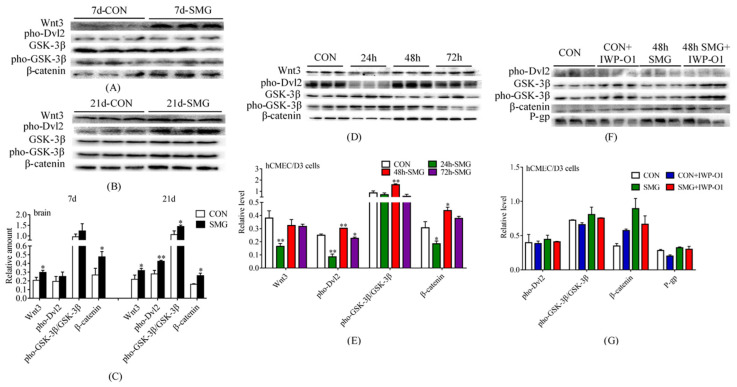
Simulated microgravity up-regulated the expression of P-gp by activating the Wnt/β-catenin signaling pathway in rat brain and hCMEC/D3 cells. (**A**–**C**) Expression levels of Wnt3, pho-Dvl2, GSK-3β, pho-GSK-3β, and β-catenin in rat brains determined using Western-Blot. (**D**–**G**) Expression levels of Wnt3, pho-Dvl2, GSK-3β, pho-GSK-3β, β-catenin, and P-gp in hCMEC/D3 cells determined using Western-Blot. The relative expression levels of proteins were expressed as the ratio of the gray value of the protein band to that of the total proteins in the same lane. The total protein gels are shown in Appendix A. * *p* < 0.05 and ** *p* < 0.01 vs. the CON group.

**Figure 7 ijms-24-05438-f007:**
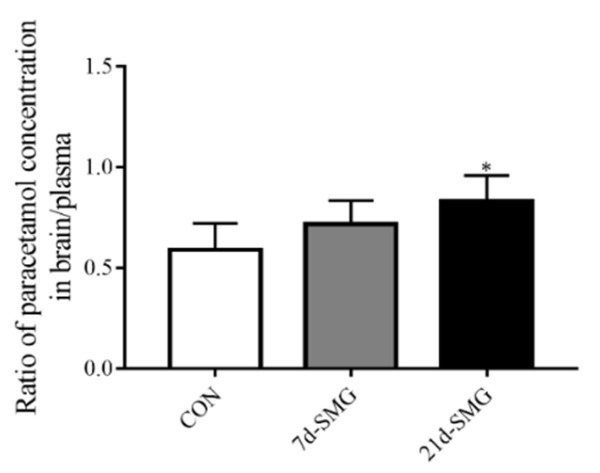
Simulated microgravity promoted the in vivo brain distribution of acetaminophen. The brain distribution of acetaminophen was reflected by the ratio of its concentration in the brain to its concentration in the plasma (*n* = 6). * *p* < 0.05 vs. the CON group.

**Table 1 ijms-24-05438-t001:** Simulated microgravity promoted the in vivo intestinal absorption of AP in rats (*n* = 6).

		Concentration (mg/mL)	*K*_a_ (10^−2^, min^−1^)	*P*_eff_ (10^−3^, cm/min)
7 days	CON	1.684 ± 0.015	7.75 ± 0.11	8.86 ± 0.16
	CON + Ver	1.516 ± 0.052 *	8.93 ± 0.37 *	10.77 ± 0.61 *
	SMG	1.571 ± 0.029 **	8.55 ± 0.21 **	10.11 ± 0.33 **
21 days	CON	1.965 ± 0.033	5.66 ± 0.19	6.01 ± 0.23
	CON + Ver	1.727 ± 0.025 **	7.44 ± 0.18 **	8.41 ± 0.25 **
	SMG	1.54 ± 0.066 **	8.76 ± 0.47 **	10.47 ± 0.76 **

Notes: * *p* < 0.05 and ** *p* < 0.01 vs. the CON group.

**Table 2 ijms-24-05438-t002:** Oligonucleotide sequences of quantitative qPCR.

Gene	Primer	Sequences
Rat *mdr1a*	Forward	5′-GGTTCGGTGCCTACTTGGTG-3′
	Reverse	5′-GATGTGGGATGCTGAGACTTTG-3′
Rat *mdr1b*	Forward	5′-GAAATAATGCTTATGAATCCCAAA-3′
	Reverse	5′-GGTTTCATGGTCGTCGTCTCTTGA-3′
Rat GAPDH	Forward	5′-TCTCTTGTGACAAAGTGGACAT-3′
	Reverse	5′-GGTGATGGGTTTCCCGTTGA-3′
Human *MDR1*	Forward	5′-TTGCTGCTTACATTCAGGTTTCA-3′
	Reverse	5′-AGCCTATCTCCTGTCGCATTA-3′
Human GAPDH	Forward	5′-ACAACTTTGGTATCGTGGAAGG-3′
	Reverse	5′-GCCATCACGCCACAGTTTC-3′

## Data Availability

The data supporting the findings reported here are available on reasonable request from the corresponding author.

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
