# Peer review of "Simulated Microgravity Alters P-Glycoprotein Efflux Function and Expression via the Wnt/β-Catenin Signaling Pathway in Rat Intestine and Brain"

_ijms, 2023, doi:10.3390/ijms24065438_

Round 1

Reviewer 1 Report

The authors have put together a very interesting paper on the role of microgravity on the specific function of the P-gp efflux pump, as evaluated in their rat animal model. The studies are well-described and rigorous. The discussion of the results is focused and does not expanded beyond what is appropriate. While the audience for this work will be relatively small at this point, the work is relevant for future studies. Merits publication. 

Author Response

Dear reviewer,

    Thank you for your careful review and affirmation of this article.

Reviewer 2 Report

The submitted manuscript “Simulated microgravity alters P-glycoprotein efflux function and expression via the Wnt/β-catenin signaling pathway in rat 3 intestine and brain” is well written and well documented.

The discussion should be extended to support the final statement of the authors in the abstract "These findings may be helpful in guiding the use of P-gp substrate drugs during spaceflight." and in the conclusions: “These results might be helpful in understanding the effects and mechanism of SMG on P-gp function and expression in the intestine and brain and provide scientific support as guidelines for the use of P-gp substrate drugs during spaceflight.”

Otherwise, the manuscript is acceptable for publication

Reviewer 3 Report

Oral drugs are used to prevent or treat the multisystem physiological damage induced by microgravity to ensure the health and safety of astronauts. Several efflux transporters located in intestinal epithelial cells and brain microvascular endothelial cells, such as permeability glycoprotein, are involved in the absorption and distribution of oral drugs. However, the microgravity-induced change in the efflux function of permeability glycoprotein and the potential mechanism have not been elucidated. Thus, the aim of this study was to investigate the alteration of permeability glycoprotein efflux function, expression, and potential signaling pathway in rats and cells under different simulated microgravity duration. The authors showed that the efflux function of permeability glycoprotein was inhibited in the rat intestine and brain and human colon adenocarcinoma cells and human cerebral microvascular endothelial cells by simulated microgravity-treated. Protein and gene expression levels of permeability glycoprotein were down-regulated in rat intestine and up-regulated in rat brain by simulated microgravity-treated. The expression was regulated by the Wnt/β-catenin signaling pathway. Furthermore, the elevated intestinal absorption and brain distribution of acetaminophen levels indicated the inhibited efflux function of permeability glycoprotein in rat intestine and brain. The authors conclude that the results might be helpful in understanding the effects and mechanism of simulated microgravity on function and expression of permeability glycoprotein in the intestine and brain and provide scientific support as guidelines for the use of its substrate drugs during spaceflight. The manuscript is well-written and the methods sound. I did not have any major concerns, only minor issues listed below:

Page 14 line 527, “In summary,”

Page 15 line 560, “In summary,”

Reviewer 4 Report

Dear Authors,

The study revealed that under microgravity efflux function of P-gp is changed that can influence the pharmacokinetics of the oral drugs during spaceflight. It is very important issue,  that touch also the multidrug resistance. From this point of view, it would be interesting to add a small discussion of how this discovery can be applied to overcome the MDR.

1.       From the Introduction it is not clear the actuality of acetaminophen metabolism and it`s efflux during the flights. How often do the astronauts use acetaminophen and what were the identified side effects? Some statistics  data should be given to confirm the actuality.

2.       Some abbreviations in the title of the sections makes it difficult to perceive information for the readers, also some ABB are absent in the appropriate ABB section

Section 4.8 : IF (should be decrypted),  it exists in ABB, but for the name of the methods it is not allowed

Section 4.9: Rat SPIP of AP  - this is disrespectful  to the reader

3.       “The rats in SMG groups were subjected to tail suspension and 30° head-down for 7 d or 21 d according to the Morey–Holton model [78]”

Why such a periods of time have been chosen? The abnormal and damaging conditions (tail suspension, 30° head-down), obviously, disrupt the work of all body systems, because not only ABC-transporters, but also other metabolic enzymes and transporters can be damaged by this condition and also vital body functions.

The model of Morey–Holton , used in this study,  thought to imitate MG: how acceptable is such a procedure with animals, did they were tail suspended for all 21 days? How humane is this for animals?

It is desirable to argument in a more wide fashion, why this model  have  been used, CON + Ver group: “CON” is absent in ABB section

4.       The main conclusions of the study is  the increased intestinal absorption and brain distribution levels of the P-gp substrates  and the inhibition of P-gp function in the intestine and brain under SMG.   What is the main recommendation for the flights? To use lower concentration of the oral drugs?  It is not clear. Also some discussion is desirable about possible physical methods to prevent MDR in cancerous patients.

5.       Rat mdr1a and Rat mdr1b primers have been used in the study. There are no explanations, what are the mdr1a and mdr1b
